# Ionomers Based on Addition and Ring Opening Metathesis Polymerized 5-phenyl-2-norbornene as a Membrane Material for Ionic Actuators

**DOI:** 10.3390/membranes12030316

**Published:** 2022-03-10

**Authors:** Oleg S. Morozov, Alexander V. Babkin, Anna V. Ivanchenko, Svetlana S. Shachneva, Sergey S. Nechausov, Dmitry A. Alentiev, Maxim V. Bermeshev, Boris A. Bulgakov, Alexey V. Kepman

**Affiliations:** 1Department of Chemistry, Lomonosov Moscow State University, 119991 Moscow, Russia; alexandr.babkin@gmail.com (A.V.B.); nuta.avi@gmail.com (A.V.I.); nechersergey@mail.ru (S.S.N.); bbulgakov@gmail.com (B.A.B.); alexkep@inumit.ru (A.V.K.); 2Faculty of Materials Science, Lomonosov Moscow State University, 119991 Moscow, Russia; shachneva20@gmail.com; 3A.V. Topchiev Institute of Petrochemical Synthesis, Russian Academy of Sciences, 119991 Moscow, Russia; d.alentiev@ips.ac.ru (D.A.A.); bmv@ips.ac.ru (M.V.B.)

**Keywords:** ion exchange membrane, ionomer, sulfonated polymer, electroactive polymer, carbon nanotube, ionic liquid

## Abstract

Two types of poly(5-phenyl-2-norbornene) were synthesized via ring opening metathesis and addition polymerization. The polymers sulfonation reaction under homogeneous conditions resulted in ionomer with high sulfonation degree up to 79% (IEC 3.36 meq/g). The prepared ionomer was characterized by DSC, GPC, ^1^H NMR and FT-IR. Polymers for electromechanical applications soluble in common polar organic solvents were obtained by replacing proton of sulfonic group with imidazolium and 1-methylimidazlium. Membranes were prepared using the above-mentioned polymers and 1-ethyl-3-methylimidazolium tetrafluoroborate (EMImBF4), as well as mixtures with polyvinylidene fluoride (PVDF). Mechanical, morphological, and conductive properties of the membranes were examined by tensile testing, SEM, and impedance spectroscopy, respectively. Dry and air-stable actuators with electrodes based on SWCNT were fabricated via hot-pressing. Actuators with membranes based on methylimidazolium containing ionomers outperformed classical bucky gel actuator and demonstrated high strain (up to 1.14%) and generated stress (up to 1.21 MPa) under low voltage of 2 V.

## 1. Introduction

Ionic electroactive polymers (EAP) are among the most promising smart materials with applicability as electromechanical transducers. Compared with the other types of EAP based actuators, ionic actuators exhibits large deformations under low applied voltages (1–5 V) [1]. Historically, the most common polyelectrolyte for preparation of ionic actuators is sulfonated tetrafluoroethylene copolymer, since the first electromechanical transducers were made using the commercially available Nafion (DuPont) [2,3]. Even though ion-polymer metal composites (IPMC) based on Nafion demonstrate high performance, there are drawbacks which limit its further development and practical applications such as fixed ion exchange capacitance (IEC) and proton conductivity, reduction mechanical and electrochemical properties at elevated temperature, low generated force, back-relaxation under direct current voltage and high cost. Many studies have been focused on the search for new polymers that can replace Nafion membranes [4,5]. Sulfonated commercially available polymers such as sulfonate polystyrene [6], sulfonated poly(ether ether ketone) [7], sulfonated poly(styrene-ran-ethylene) [8], sulfonated poly(styrene-b-ethylene-co-butylene-b-styrene) [9,10,11], sulfonated polyetherimide [12], sulfonated polyimide [13] sulfonated styrenic pentablock copolymer [14,15,16], sulfonated polyphenylsulfone [17], sulfonated poly(1,4-phenylene ether-ether-sulfone) [18] were employed to prepare ionic actuators. IPMC actuators based on these polymers could operate only in the swollen state. The electrolyte is usually water or an aqueous salt solution. The main disadvantage of actuators with aqueous electrolyte is their unsuitability for prolonged operation in the open air. Both evaporation and electrolysis due to a narrow electrochemical window of water quickly and significantly reduce the efficiency of electromechanical devices. Two approaches were proposed to solve this problem: the use of non-volatile, stable electrolytes and encapsulation of actuators. Although encapsulated actuators are able to work in air for a long time [19,20,21], this approach does not solve the problem of the electrochemical stability of water. The capsule also increases the total stiffness of the actuator, which reduces their efficiency. Another approach is to replace water with room temperature ionic liquids (IL) as an electrolyte [22,23,24,25]. ILs have unique and advantageous properties such as high ionic conductivity, thermal stability, wide electrochemical window, and immeasurably low vapor pressure. Polymer gel electrolytes based on ILs demonstrate conductivity at a room temperature above 10^−3^ S/cm, which is enough for practical use [26]. High efficiency of carbon nanotubes-based electrodes (bucky gel electrodes) in actuators comprising ionic liquid was shown by Asaka [25,27,28]. On the one hand, membranes based on ionomers demonstrate conductivity significantly lower than pure electrolytes, on the other hand, the appearance of additional mobile cations increases the deformation of actuators based on such membranes [29]. Watanabe described high performance printable polymer actuators based on soluble sulfonated polyimide comprising ionic liquid with carbon electrodes [30]. More recently, fast response bucky gel actuator based on specially synthesized sulfonated block copolymer containing IL was described [31,32]. The most common ionic liquid for actuators based on inomers is 1-ethyl-3-methylimidazolium tetrafluoroborate (EMImBF4), and the actuators typically show a deformation of 0.4–0.6% at a voltage of 1–3 V.

Highly conductive sulfonated polymer based polynorbornene prepared using ring-opening metathesis polymerization (ROMP) for proton exchange membrane was described by Ramani [33]. This polymer was synthesized by co-polymerization of norbornene substituted benzenesulfonyl chloride, norborbene and dicyclopentadiene and subsequent hydrolysis to form an ionomer. In this study, we describe the preparation of poly(phenyl norbornenes) synthesized by two polymerization routes, namely, ROMP and addition polymerization [34], their sulfonation under homogeneous conditions, and their use as electrolyte membranes of ionic actuators. Homogeneous sulfonation of the polymer in different chlorinated solvents under mild conditions by propionyl sulfate was studied. The polymers were characterized by ^1^H NMR, FTIR, GPC and DSC. For electromechanical applications cation of the sulfonated polymer was exchanged by imidazolium and 1-methylimidazolium. The resulting polymers were blended with polyvinylidene fluoride (PVDF) and ionic liquid to achieve polyelectrolyte membranes. Finally, bucky gel actuators were fabricated and tested in terms of deformation and blocking force under 2 V DC and compared with classical counterpart.

## 2. Materials and Methods

### 2.1. Materials

Unless otherwise stated, all manipulations were carried out using standard Schlenk techniques under an argon atmosphere. 5-phenyl-2-norbornene was synthesized by the Diels-Alder reaction from styrene and dicyclopentadiene using standard procedure [35]. Toluene, o-xylene was refluxed over Na and distilled in an argon atmosphere, stored over sodium wire. Dichloromethane, 1,2-dichloroethane and chloroform was distilled over CaH_2_ in an argon atmosphere prior to use. Methanol, 2,2′-methylenebis (6-tert-butyl-4-methylphenol), p-toluenesulfonyl hydrazide, the first-generation Grubbs catalyst, Pd(OAc)_2_, tricyclohexylphosphine, propionic anhydride, imidazole, 1-methylimidazole, ethyl bromide, sodium tetrafluoroborate, sodium tetrakis [3,5-bis(trifluoromethyl)phenyl]borate (NaBAr^F^) were purchased from commercial sources (Acros) and were used without prior purification. PVDF purchased from the Konstantinov Kirovo-Chepetsk Chemical Combine as Fluoroplast-2. Single wall carbon nanotubes (SWCNT) purchased from the OCSiAl. 1-Ethyl-3-methylimidazolium tetrafluoroborate (EMImBF_4_) was synthesized by literature procedure [36].

### 2.2. Polymers Characterization

All NMR spectra were acquired at 600 MHz (^1^H) on a Bruker Avance III Ultrashield spectrometer. NMR spectra were recorded in deuterated chloroform for non-ionic compounds and in deuterated dimethyl sulfoxide for sulfonated polymers. Chemical shifts were referenced to residual solvent signals. Fourier transform infrared (FTIR) spectra were acquired in the range of 4000–400 cm^−1^ on Bruker Tensor-27 spectrophotometer using KBr pellets. The differential scanning calorimetry (TA Instruments Q20 V24.11) data were obtained in a sealed aluminum pan with a heating rate of 10 °C/min under N_2_ purge. The molecular weights of the polymers were evaluated by gel permeation chromatography (GPC) on a Waters high pressure chromatograph (Microgel mix column 1–5 μm 500 × 7.7 mm Chrompack, sample volume was 100 μL, sample concentration was 1–2 mg/mL in chloroform) equipped with a refractometric detector. Calibration was performed according to polystyrene standards with molecular weights of 1 × 10^3^ ÷ 1 × 10^6^. The calculation of molecular mass characteristics was performed according to the calibration dependence, which is linear in the range 10^3^ ÷ 10^6^. Some characterization of sulfonated polymer prepared via ROMP is published previously as raw data without discussion [37].

### 2.3. Polymerization of 5-phenyl-2-norbornene (ROMP)

The procedure is described on the example of an experiment with a monomer/catalyst ratio of 3000/1 and an initial monomer concentration in the reaction mixture of 0.28 M. Other experiments were carried out similarly. A solution of the first-generation Grubbs catalyst (1.0 × 10^−3^ M) in dry toluene was prepared immediately before polymerization. 1.00 g (5.9·mmol) of 5-phenyl-2-norbornene and 18 mL of dry toluene were introduced into a vial under argon atmosphere. Polymerization was initiated by adding 2.00 mL (2.0 × 10^−6^ mol) of the catalyst solution. Stirring was continued for 2 h. Each time the reaction mixture became so viscous that stirring was difficult, it was diluted with 5.0 mL of dry toluene (total volume: 40.0 mL). Polymerization was terminated by the addition of vinyl ethyl ether. The polymer was precipitated into methanol containing an oxidation inhibitor (2,2′-methylene bis (6-tert-butyl-4-methylphenol)). Then the polymer was filtered, washed 3 times with methanol and dried. The polymer was re-precipitated twice from toluene to methanol and dried under vacuum at 40 °C to constant weight. Yield: 0.88 g (88%). Mw = 1.2 × 10^6^, Mw/Mn = 3.5.

^1^H NMR (δ, ppm, CDCl_3_): 7.40–6.71 (m., 5H, Ar−H), 5.67–4.44 (m., 2H, −HC=CH−), 3.42–0.72 (m., 7H).

### 2.4. Synthesis of Hydrogenated Poly(5-phenyl-2-norbornene) HPPNB

To a two-neck flask (1 L) equipped with a reflux condenser and a magnetic stirrer 15 g (88 mmol) of 5-phenyl-2-norbornene and 300 mL of dry xylene were added. Polymerization was initiated by adding a solution of 24·mg (2.9 × 10^−5^ mol) of the first-generation Grubbs catalyst in 3 mL of xylene. Stirring was continued for 1 h. Each time the reaction mixture became so viscous that stirring was difficult, it was diluted with 50 mL of dry xylene (total volume: 200 mL). Vinyl ethyl ether was added to reaction mixture to terminate polymerization. Then 66 g (0.35 mol) of p-toluenesulfonyl hydrazide was added to the reaction mixture. The mixture was boiled under stirring for 4 h. The resulting viscous polymer solution with unreacted p-toluenesulfonyl hydrazide was decanted and the product was precipitated into methanol. The product was filtered, washed 3 times with methanol and dried under vacuum. The polymer was re-precipitated twice from toluene to methanol and dried under vacuum at 40 °C to constant weight. Yield: 13.2 g (87%). Mw = 8.2 × 10^5^, Mw/Mn = 3.5.

^1^H NMR (δ, ppm, CDCl_3_): 7.44–6.74 (m., 5H, Ar−H), 3.38–0.27 (m., 11H).

### 2.5. Polymerization of 5-phenyl-2-norbornene (Addition Polymerization)

The polymerization was carried out at a 5-phenyl-2-norbornene/Pd(OAc)_2_/NaBAr^F^/PCy_3_ ratio of 3000/1/5/2 and an initial monomer concentration in the reaction mixture of 2.4 M. In a glovebox under argon atmosphere 6.8 mL of Pd(OAc)_2_ (0.01 M, 0.068 mmol) in dry chloroform, 6.8 mL of NaBARF (0.05 M, 0.34 mmol) in dry chloroform were mixed in a vial and 1 drop absolute methanol was added then 6.8 mL of a solution of PCy3 (0.02 M, 0.136 mmol) in dry chloroform and the resulting solution was stirred for 5 min. In a separate 500 mL flask 10.2 g of 5-phenyl-2-norbornene (0.06 mol) was dissolved in 9 mL of dry chloroform. The polymerization was initiated by adding 6.0 mL of the catalytic mixture (0.02 mmol Pd) to the flask with the monomer under stirring. Stirring was continued for another 20 h. As the viscosity increased, the reaction mixture was diluted stepwise with dry chloroform (the total volume of the mixture was 160 mL). The reaction mixture was precipitated into methanol, the polymer was washed 3 times with methanol and dried in vacuo. Then the polymer was twice reprecipitated from toluene into methanol and dried in vacuum at 80 °C to constant weight. Yield: 6.1 g (60%). Mw = 3.1·10^5^, Mw/Mn = 1.4.

^1^H NMR (δ, ppm, CDCl_3_): 8.37–6.07 (m., 5H, Ar–H), 3.38–1.27 (m., 11H).

### 2.6. Copolymerization of 5-phenyl-2-norbornene and 5-docecyl-2-norbornene (Addition Polymerization)

The copolymer was synthesized by a procedure similar to that described above with the same ratio of catalyst and monomers. The procedure was repeated twice with a molar ration of 5-phenyl-2-norbornene to *5-docecyl-2-norbornene* 10/1.3 (total mass 2.05 g) and 10/2.6 (total mass 2.20 g). The reaction mixture was stirred for 5 h instead of 20 h and was diluted 4.5 times. Yield: 0.86 (42%), Mw = 1.17 × 10^6^, Mw/Mn = 1.69, monomer ratio 27/73 (Dodecyl/Ph); 0.77 g (35%), Mw = 2.88 × 10^6^, Mw/Mn = 2.51 monomer ratio 1/1 (Dodecyl/Ph).

^1^H NMR (δ, ppm, CDCl_3_): 8.07–6.20 (m., Ar−H), 3.34–0.57 (m., aliphatic).

### 2.7. Sulfonation Procedure

3.00 g of polymer was dissolved in 60 mL of chlorinated solvent (dichloromethane, chloroform or 1,2-dichloroethane) under argon in an oil bath heated at 40 °C. After the polymer was dissolved, in a separate flask propionic anhydride (9.52 g, 4.2 eq.) was added to 10 mL of solvent cooled to 0 °C. To the solution 6.82 g (4.0 eq.) of 98% sulfuric acid was added dropwise at such a rate that the temperature of the solution did not exceed 5 °C. The sulfonating mixture was added to the solution of the polymer and the mixture was stirred under argon at 40 °C for 3 h. Immediately after addition of propionyl sulfate solution, the reaction mixture changed color to dark brown. After 3 h, the gelatinous reaction mixture was poured into 50 mL of 2-propanol to terminate sulfonation. All volatiles were isolated on rotary evaporator under vacuum at 60 °C. The residue was transferred to a flask contains dimethyl sulfoxide (50 mL). The mixture was stirred at 150 °C under reduced pressure (400–500 mm Hg) to remove remaining solvents. After 4 h clear brown solution was obtained. The resulting solution was cooled to room temperature and poured into 300 mL of diethyl ether. The precipitated product was filtered and washed 3 times with diethyl ether and dried under vacuum at 60 °C for 12 h.

SHPPNB: ^1^H NMR (δ, ppm, DMSO-d6):7.75–7.38 (m., otho 2H of Ph-SO_3_H), 7.32–6.87 (m., 5H of Ph + meta 2H of Ph-SO_3_H), 3.71–3.11 (m., 1H), 2.21–0.26 (m., 10H).

SAPPNB: ^1^H NMR (δ, ppm, DMSO-d6):8.35–8.10 (m., otho 2H of Ph-SO_3_H), 7.76–7.38 (m., 2H of Ph-SO_3_H), 7.37–6.96 (m.,1.32H of Ph), 3.78–3.56 (m., 1H), 2.41–0.86 (m., 8H).

### 2.8. Synthesis of Polymer with Imidazolium and 1-methylimidazolium Cation

1.00 g of sulfonated hydrogenated poly(5-phenyl-2-norbornene) was added to 25 mL of dimethylformamide. The mixture was stirred at 120 °C 1 h. To the resulting gelatinous mass 270 mg of imidazole or 325 mg of 1-methylimidazole was added. The reaction mixture was stirred until it became homogeneous and then it was cooled to room temperature. The clear brown solution was poured into a 400 mL beaker containing 250 mL of diethyl ether. After filtration and washing with 2 × 50 mL of diethyl ether the product was dissolved in 30 mL of methanol. The solution was poured again into tetrahydrofuran. In addition, the procedure was repeated once more time. Polymer precipitate was dried under vacuum.

SPPhNB-MIm: ^1^H NMR (δ, ppm, DMSO-d6): 8.91 (br. s, 1 H, NCHN),7.66–7.42 (m., otho 2 H of Ph-SO_3_H + CH=CH of MIm), 7.28–6.89 (m., 5 H of Ph + meta 2 H of Ph-SO_3_MImH^+^), 3.82 (s, 3 H), 3.40–2.82 (m., 1 H), 2.24–0.29 (m., 10 H).

SPPhNB-Im: ^1^H NMR (δ, ppm, DMSO-d6): 8.58 (br. s, 1 H, NCHN),7.63–7.32 (m., otho 2 H of Ph-SO_3_H + CH=CH of Im), 7.30–6.78 (m., 5 H of Ph + meta 2 H of Ph-SO_3_MImH^+^), 3.38–2.73 (m., 1H), 2.21–0.27 (m., 10H).

SAPhNB-MIm: ^1^H NMR (δ, ppm, DMSO-d6): 8.86 (br. s, 1 H, NCHN), 8.53–8.06 (m., otho 2 H of Ph-SO_3_H), 7.76–6.75 (m., 5 H of Ph + meta 2 H of Ph-SO_3_MImH^+^+ CH=CH of MIm as two narrow singlets at 7.57 and 7.51 ppm), 3.79 (s, 3 H, CH_3_ of MImH^+^), 3.61–3.28 (m., 1 H), 2.35–0.84 (m., 8 H).

SAPhNB-Im: ^1^H NMR (δ, ppm, DMSO-d6): 8.53 (br. s, 1 H, NCHN), 8.35–8.08 (m., ortho 2 H of Ph-SO_3_MImH^+^), 7.84–6.85 (m., 5 H of Ph + meta 2 H of Ph-SO_3_H + CH=CH of Im as a narrow singlet at 7.40 ppm), 3.72–3.55 (m., 1 H), 2.41–0.86 (m., 8 H).

### 2.9. Degree of Sulfonation Measurement

The sulfonation degree was determined by reverse titration method. To a 100 mL flask 100–200 mg of polymer was weighed and 10 mL of methanol, 10 mL of 0.1 M of NaOH and phenolphthalein were added. The mixture was stirred for 24 h and titrated with 0.1 M HCl. The procedure was repeated triple times and ion-exchange capacity (IEC) was calculated using average acid amount.

### 2.10. Membrane Preparation

The polymer membranes were prepared by casting method as follows. The casting solutions were prepared by dissolving 750 mg of ionomer and 750 mg of EMImBF_4_ or by 750 mg of PVDF, 375 mg of ionic polymer and 375 mg of EMImBF_4_ in DMF at 100 °C on a hot plate at 1400 rpm for 6 h. The mixtures were degassed under vacuum prior to casting. Resulting homogeneous solutions were cast on a Petri dishes (Ø 105 mm) and left to dry in oven at 80 °C for 12 h. Then the films were stripped from the dishes and weighted to control full removal of the solvent.

### 2.11. Membrane Characterization

The ionic conductivity of the polymer membranes was determined by the ac complex impedance technique over the frequency range from 0.1 Hz to 5 MHz using a P-45X potentiostat/galvanostat equipped with FRA-24M module (Electrochemical Instruments). The samples were sandwiched between symmetrical cells containing two coin-shaped steel electrodes with area of A (0.25 cm^2^) at the open circuit potential with a small amplitude ac voltage of 5 mV to measure membrane impedance, Z (Ω). The thickness of the sample L (cm) was measured with a micrometer. The conductivity (σ, S/cm) was then calculated from the equation: σ = L/(Z × A). Scanning electron microscopy (SEM) was performed on an TESCAN Vega 3 instrument with an accelerating voltage of 20 kV. The samples were sputter-coated with approximately 10 nm of gold before analysis. The membranes were fractured in liquid nitrogen to observe the cross-section morphology. Mechanical properties of the membranes were measured with the Universal Testing Machine (Instron 5985) at room temperature. The membranes were cut into small pieces (~50 mm × 9 mm) for testing. The exact width and thickness were measured with a caliper and micrometer, respectively, for each specimen and these values were used for strength calculations.

### 2.12. Preparation of Actuators

Bucky-gel actuators were prepared similar to the method previously reported [27]. In an agate mortar, 100 mg of an ionic liquid (EMImBF4) was added to 40 mg of carbon nanotubes. The mixture was triturated and 0.5 mL of DMF was added and ground again. The operation was repeated three more times. The resulting gel was added to a solution of 60 mg of PVDF in 4 mL of DMF and the mixture was placed in an ultrasonic bath for 12 h at 50 °C. The resulting thick paste was transferred into a Petri dish (Ø 95 mm) and 12 mL of DMF was added. The mixture was evenly distributed in shape, either using an ultrasonic bath or using a spatula. The dish was placed in an oven for 3 h (100 °C). The thickness of the obtained electrode film was 15–20 μm. Samples of rectangular shape of membrane (~65 × 6 mm) and two electrodes (~60 × 5 mm) were cut. The polymer membranes were sandwiched between two bucky-gel electrodes via hot pressing at 120 °C. Then the actuators were cut to 60 × 5 mm for tip displacement and blocking force tests.

### 2.13. Characterization of Actuators

The actuator strip was connected to the glassy carbon electrodes over graph paper and the displacement was registered by digital camera. The constant and alternating square-wave voltages were applied to the actuator strip by a P-45X potentiostat/galvanostat equipped with FRA-24M module (Electrochemical Instruments). Constant voltage +3 V and alternating voltages ±2 V with 0.1, 0.05, 0.025 and 0.0125 Hz were used. The voltage and current were monitored simultaneously with a software ES8. The actuator strip showed a bending motion when the voltage was applied. The strain is calculated from the displacement [38] by,
(1)$$\varepsilon =\frac{2\mathrm{Dh}}{L^{2}{+D}^{2}}\times 100\%$$
where D—displacement, L—free length, h—thickness.

The blocking force was measured using analytical balances similar to the method described elsewhere [39]. The maximum generated stress (σ) during actuation motion was calculated by using two parameters, the maximum strain (ε_max_) and the Young’s modulus (Y_el_) of electrodes, according to Hooke’s law: σ = εY_el_. The mechanical stress was also calculated according to the equation σ = 6Fl/bh^2^. where F is the blocking force generated by the sample, l, b and h are its length, width and thickness, respectively.

## 3. Results

### 3.1. Polymer Synthesis and Characterization

Norbornene derivatives are active in ring-opening metathesis polymerization (ROMP) in the presence of the first-generation Grubbs catalyst [40] as well as in addition polymerization using palladium catalyst [34] Two types of polymers were successfully obtained based on 5-phenyl-2-norbornene (Figure 1). The structures of the formed polymers were confirmed by 1H NMR spectroscopy (for spectra see Appendix A).

Variation of metathesis polymerization conditions (Table 1) revealed several effects. First, as expected, with an increase in the monomer/catalyst ratio, the yield of polymerization decreases, and the molecular weight of the polymer increases. Secondly, an increase in the initial monomer concentration in the reaction mixture leads to a decrease in molecular weight and an increase in the polydispersity index of the polymer. This is due to the fact that in concentrated solutions there may be difficulties in diffusing the monomer to the active polymerization center, due to their high viscosity. Guided by the data obtained, we found the optimal conditions for the synthesis of the polymer.

Reduction of the chain double bonds of the polymer was carried out directly after polymerization without poly(5-phenylnorbornene) isolation and purification steps, by boiling the polymer solution with p- toluenesulfonyl hydrazide. As a result, a polymer with a saturated main chain and side phenyl groups was formed, which was confirmed by ^1^H NMR spectroscopy (for spectrum see Appendix A). It is interesting to note that GPC analysis showed a significantly lower molecular weight of the hydrogenated polymer of 820 × 10^3^ g/mol than for PPhNB. This can be explained by the fact that the GPC method can only determine relative molecular weights, i.e., molecular weight polystyrene, having in the solution of the same size of the coil, as the polymer under study. PPhNB and hydrogenated polymer have different chain stiffness, macromolecular coils of samples of these polymers with the same degree of polymerization will have different sizes and, accordingly, different relative molecular weights. However, PPhNB and HPPhNB have the same *M*_w_/*M*_n_.

In general, additive polymers are characterized by greater thermal stability, higher glass transition temperature, and, most importantly, lower solubility compared to polymers obtained by metathesis. Bulk polymerization of the substituted norbornenes requires the use of active catalysts based on late transition metals such as Pd or Ni [41]. To obtain a soluble polymer based on 5-phenyl-2-norbornene, a stepwise dilution of the reaction mixture and a very active catalyst were used, since a polymer with too high molecular weight is formed at a low monomer concentration. A cationic complex of palladium with PCy_3_ ligand was used as a catalyst in an amount of 0.033 mol.%. A soluble polymer was obtained with a yield of 60% and Mw 3.1 × 10^5^ g/mol with a total tenfold dilution of the reaction mixture in 20 h. Under similar conditions, a block copolymer of 5-phenyl-2-norbornene and 5-dodecyl-2-norbornene was also obtained in 42% yield. According to ^1^H NMR data, in both structures there were no hydrogen signals at the double bond of norbornene.

In the initial experiments, the sulfonation conditions were selected using HPPhNB as the substrate. Sulfonation reaction was carried out by propionic sulfate under homogeneous condition similar to that described for polystyrene in different chlorinated solvents [42]. Dichloromethane, chloroform and 1,2-dichloroethane were tested as solvents. In all cases initial polymer dissolved completely within 3 h. Gel formation was observed in all the solvents, which makes further reaction difficult. Gelation time was determined by stopping stirrer bars (see Table 2). Sulfonation degree was determined by titration. All polymers were not soluble in water and the polymers obtained in dichloromethane was soluble only in DMSO. To increase sulfonation degree the reaction with four instead of two equivalents of propionyl sulfate was carried out. For further study the polymer with maximum sulfonation degree was chosen.

Study on sulfonation of APPhNB showed similar results: four equivalents of sulfonating agent CH_2_Cl_2_ allowed to obtain a polymer with higher (81%) sulfonation degree. Inversely, SAPPhNB occurred to be soluble in H_2_O and MeOH even better than in high boiling polar solvents such as DMF and DMSO. Sulfonation of block-copolymer was conducted in CHCl_3_ due to better solubility of polymer. Complete dissolution of the polymer took about a day, while the resulting solution looked much more viscous compared to APPhNB one. The sulfonated polymer occurred to be totally insoluble: even 0.5 mg of the product putted into NMR tube with DMSO-d6 for 3 days showed a blank spectrum. Therefore, all further work was carried out only with SHPPhNB and SAPPhN.

The DSC curves for sulfonated and initial HPPhNB and APPhNB are represented on Figure 1. As expected, the absence of melting peaks on curves indicates that the polymers are amorphous. Sulfonation leads to increase in glass transition temperature of polymers in both cases.

Replacing proton of sulfonate group with organic cations should lead to increase of solubility and electrochemical stability of the polymer. For membrane preparation the polymer was modified by adding imidazole and 1-imidazole (Figure 2). The cation exchange reaction was carried out in DMF and MeOH.

The structures of the polymers were confirmed by FTIR (Figure 2) method. The bands around 2925 and 2850 cm^−1^ related to the C–H stretching vibration of the CH_2_ on the main chain of polymers were observed in all the samples and were used as an internal standard to normalize all the spectra [43]. The peak around 752 cm^−1^ is characteristic of the out-of-plane bending vibration of C-H groups in the substituted benzene ring [44]. The intensity of absorbance for the in-plane C=C skeletal vibration of the phenyl ring (1492 cm^−1^) was reduced with sulfonating of the polymer [40]. The appearance of the bands at 1130 cm^−1^ after sulfonation can be attributed to the symmetric vibration of the sulfonic group. In the same time in the spectra of polymers with imidazole and 1-methylimidazole additional bands of symmetric stretching vibration of −SO_3_^−^ group around 1223 cm^−1^ were revealed [45]. The absorption band at 1665 cm^−1^ indicates protonated form of imidazole and 1-metylimidazole [46]. These results approved that −SO_3_H groups deprotonate with imidazole and methylimidazole to form −SO_3_^−^ groups.

The ^1^H NMR spectra of the polymers are presented in Figure 3. After sulfonation, two new signals appear, corresponding to ortho (with a maximum at 7.50 and 8.24 ppm) and meta (with a maximum at 7.20 and 7.55 ppm) protons of Ph-SO_3_H. After the addition of imidazole and methyl imidazole, the N-CH-N signals appear at ~8.5–8.6 and ~8.9 ppm, respectively. The signals of the remaining two protons of the imidazole ring overlap with the signals of ortho protons of Ph-SO_3_H. It should be noted that the protons of the meta position are characterized by a high value of the relaxation time, therefore, the integrated intensity of this signal is underestimated. To avoid this, the NMR spectra were registered with 40 s delay between pulses. Sulfonation degree of the polymer can be determined from the NMR spectra as follows. SD = 1 − (I − 2)/5, where I is the integrated intensity of the overlapping signals of meta protons of Ph-SO_3_X (2H) and non-sulfonated phenyl group (5H). The sulfonation degrees obtained from the NMR spectra for SNPPhNB, SNPPhNB-Im, SNPPhNB-MIm are 77, 76, 77% and for SAPPhNB, SAPPhNB-Im, SAPPhNB-MIm are 79, 78 and 79%, respectively. The difference between values obtained from NMR spectra and titration may be since a small amount of sulfuric or propionic acid (<0.5%) remains in the polymer sample after sulfonation. In any case, in the process of obtaining polymers with imidazolium cations, these residual amounts of acids were removed.

### 3.2. Membrane Preparation and Characterization

Initially, films of SHPPhNB and SAPPhNB were prepared by dry casting method from solution in DMSO and MeOH, respectively. These attempts to manufacture membranes were unsuccessful. The films turned out to be too fragile and crumbled when removed from the substrate. Similarly, it was not possible to obtain SAPPhNB-Im and SAPPhNB-MIm films. SHPPhNB-Im, SHPPhNB-MIm membranes were successfully prepared from DMF solutions. Next, films of polymers with an ionic liquid were made. The ionic liquid acts as a plasticizer and serves to increase the ionic conductivity of the membranes. All membranes were found to be insufficiently strong for stretching tests, especially SAPPhNB/EMImBF4, which was not suitable for testing even for conductivity. The ionic conductivities for the rest membranes are presented in Table 3.

The results presented in Table 3 show a significant effect of the cation on the ionic conductivity of the membranes. This effect is especially noticeable for membranes made from the neat polymers. All membranes containing ionic liquid exhibit higher ionic conductivities. It is known that with an increase in the fraction of ionic polymers in mixtures with an ionic liquid, the membrane conductivity decreases due to formation of ion complex between SO_3_H group and imidazolium cation [29,30]. Although the introduction of an imidazolium cation significantly reduces this effect. On the other hand, all membranes were prepared with the same mass, not molar, ratio of a polymer to ionic liquid. To account for this difference, the molar ratios were obtained from NMR spectra and collected in Table 3. It can be seen that it is not the ionic liquid content, but rather the chemical nature of the polymers determines the ionic conductivity to a greater extent. The SHPPhNB polymer, in contrast to the SAPPhNB, has flexible CH_2_CH_2_ bridges in its structure, which probably leads to the formation of ion channels and increases the ionic conductivity.

To obtain a strong and stable membrane, the polymer electrolyte gel can be placed into the matrix of another non-ionic fluoropolymer [29,47,48]. Membranes based on the obtained polymers in PVDF matrix were successfully manufactured. All the were composed in PVDF/ionic polymer/EMImBF_4_ 50/25/25 wt.% ratios. The mechanical properties of the membranes obtained by stretching, as well as the ionic conductivity, are presented in Table 4 (for strain-stress curves see Appendix A).

The membranes obtained based on SHPPhNB-MIm demonstrated the highest mechanical properties in terms of fracture strain and tensile strength among the considered samples. At the same time, a membrane with SAPPhNB-MIm is characterized by higher value of the elastic modulus, which is a more important parameter in electromechanical devices, since during actuation bending strain of electrolyte membrane does not exceed 1%. Film based on SAPPhNB-Im demonstrated extremely low conductivity (~10^−5^–10^−6^ mS/cm) and was not tested for mechanical properties. Interestingly, membranes based on imidazolium and 1-methylimidazolium differ significantly in both mechanical properties and ionic conductivity.

The reason for these differences is the morphology of the membranes. Figure 4 shows SEM images of upper and lower surfaces and the cross-sections of the membranes. In the case of SHPPhNB, a sponge-like structure of PVDF is observed; sulfonated polymers occupy free space in the pores. On the contrary, spherulite structure of PVDF is observed in case of SAPPhNB. Both membranes containing imidazolium have an asymmetrical structure. The concentration of SHPPhNB-Im increases to the bottom of the film, and the top surface is enriched with PVDF. An inverse distribution of PVDF and SAPPhNB-Im is observed: a dense PVDF film is formed on the lower surface. This explains the low ionic conductivity of the membrane. In methylimidazolium based membranes, the polymers are uniformly distributed both on both surfaces and in the thickness of the membranes. The distribution of SHPPhNB-MIm in the membrane was determined by EDX-sulfur mapping (image is available in Appendix A).

### 3.3. Bucky-Gel Actuators Fabrication and Performance

Bucky gel actuators were made via hot-pressing according to slightly modified literature procedure [27]. According to the theory of bending of a three-layer bucky-gel actuator [25], the rigidity of the electrolyte layer has a slight effect on the deformation of the actuator, and generated force depends only on the Young’s modulus of electrodes. The mechanical properties of used electrodes are presented in Appendix A. The thickness of the actuator, and especially the thickness of the electrodes, significantly affect both maximum tip displacement and the blocking force. On the other hand, membranes based on ionomers provide additional mobile cations, which can significantly affect the deformation of the actuator. To compare the efficiency of the membranes, actuators of the same size (60 × 5 mm) using the same electrodes were fabricated. A standard bucky gel actuator was also made, the membrane of which consisted of PVDF and ionic liquid in mass ratio of 50/50. Initially, actuators were tested at 2 V to determine the maximum deformation in stationary mode. In all cases, bending occurred towards the anode. Since the geometric dimensions, especially the thickness of the actuators, were slightly different, the strain values were calculated for a more accurate comparison of their performance. The diagrams on Figure 5 show the time dependence of strain for two groups of actuators. The first group includes actuators with membranes based on ionic polymers and ionic liquid (Figure 5), and the second group includes actuators with PVDF blend membranes (Figure 5). All actuators with ionic polymers are superior to the standard bucky gel actuator in maximum strain. Actuators with methyl-imidazolium cation demonstrate greater performance compared to imidazolium counterpart both in the case of pure polymers and in the case of PVDF blends. The bending speed correlates with ionic conductivity. It is worth noting that after turning off the voltage, all actuators retain their position.

The maximum strain at different voltage for all actuators are collected in Table 5. Maximum strain of previously reported in [29] actuator based on Nafion ionomer membrane with the same electrolyte EMImBF4 and similar bucky gel electrodes is also presented for comparison.

Under square wave alternating voltage (±2 V) all actuators except SHPPhNB-Im exhibit a symmetric bend with a change in polarity. This can be explained by differences in membrane morphology. Photographs of the actuators based on SHPPhNB and standard bucky gel actuator at different time under ±2 V are presented in Appendix A. The shape of the sample during deformation should be an arc of a circle, and any deviation from this shape indicates an uneven distribution of mechanical or electrical properties along the length of the sample. In addition, a large resistance of the electrodes leads to decrease of the voltage by the end of the actuator and as a result unevenly deformation occurs. Therefore, the form of deformation is a good indicator of the quality of the actuator. Only the standard actuator showed deviations from the arc shape, probably due to poor adhesion between electrodes and PVDF membrane. It worth also noting electrolyte leakage was only observed when testing a standard PVDF/IL actuator. Maximum generated strains of the actuators based on ionomers at different frequencies are presented in Figure 6. Additionally, the blocking force of the actuators was measured. In Figure 6, the dependence of blocking force on time are presented. In all cases a slight decrease (2–4%) in efficiency was found when measuring the blocking force. This is probably due to Joule heating, which leads to a decrease in the stiffness of the actuators.

The graphs shows that the rate of force generation, similarly to the strain rate, depends on ionic conductivity. It is worth noting that the values of the force cannot be directly compared due to the different thicknesses of the actuators. For a more accurate comparison of the samples, the mechanical stress values were calculated according to the cantilever bending equation and Hooke’s law. The results are presented in Table 6. Both types of actuators based on methylimidazolium outperform the imidazolium counterpart and classical bucky gel actuator.

The actuator fabricated with sulfonated hydrogenated poly(phenyl norbornene) exhibited a competitive electromechanical performance compared with recently reported low-voltage bucky gel actuators [49]. However, it has the main disadvantage of all transducers based on ionic liquids, which is the relatively low response time. Therefore, our further studies are aimed at modifying the electrodes and searching for an ion-conducting medium that ensures fast operation of the drives.

## 4. Conclusions

A highly sulfonated ionic polymers were obtained under homogeneous conditions by sulfonation of two types of poly(phenyl norbornene) prepared via ring opening metathesis and addition polymerization. After sulfonation, the glass transition temperature of the polymers increased. The polymer prepared via ROMP was hydrophobic and soluble only in DMSO under heating. In opposite addition polymer was hydrophobic and soluble in common organic solvents. To increase solubility and general volume of mobile cations polymers were modified by replacing of protons of sulfo-groups by imidazolium (Im) and methylimidazolium (MIm). The two types of membranes were fabricated. The first type membranes contain ionomers and ionic liquid (EMImBF_4_). The second type is the blend of non-ionic fluoropolymer (PVDF), ionomer and EMImBF_4_. A significant effect of the cation on the ionic conductivity of the membranes was observed. SEM showed an asymmetric structure of membrane with imidazolium cation and uniformly distributed ionic polymers in PVDF. Bucky gel actuators fabricated with the membranes demonstrated stable performance under air conditions. The actuators prepared with MIm outperform Im based counterpart and classical bucky gel actuator in both bending strain and generated force under 2 V. As a result, sulfonated poly(phenyl norbornene) can become a promising platform for the manufacture of polyelectrolyte membranes and ionic gels for a wide range of applications due to its stability, good solubility and high IEC (3.36 meq/g).

## Data Availability

Some raw spectral, thermal, and electrochemical data are available https://data.mendeley.com/datasets/wcx58k2rp2/1 (accessed on 20 January 2022). The rest data presented in this study are available on request from the corresponding author.

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
