# Peer review of "Ionomers Based on Addition and Ring Opening Metathesis Polymerized 5-phenyl-2-norbornene as a Membrane Material for Ionic Actuators"

_membranes, 2022, doi:10.3390/membranes12030316_

Round 1

Reviewer 1 Report

This manuscript discribes a new PVDF based composite membrane for ionic actuators. The manuscript may be suitable for this journal after revision.

  1. What is the most important performance for ionic actuator, conductivity or else? You should show the data in the conclusion and abstract section.
  2. In introduction section, the review on performance data for ionic actuators should be analyzed in details.
  3. What is the important improvement? You mentioned “generated stress (up to 1.21 MPa) under low voltage of 2V”. Please provide some refs data as compare.
  4. The english writing should be improved. There are some mistakes, such as “he membranes”Line 17 Page 1, “exhFibits” Line 31 Page 1, ....

Author Response

Thank you very much for your valuable comments and feedback regarding our research paper.

1. You ask a very correct, but also a very difficult question. The topic of ionic actuators is rather new, and to date, there is no clear methodology for describing their properties in the literature. In general, it can be said that the efficiency of actuators is described in terms of their bending, generated force (blocking force), strain rate and operating stress. In various articles, either the displacement of the end of the actuator or the deformation is used to describe the bending. Obviously, the displacement depends not only on the materials of the actuator, but also on its geometric dimensions (primarily thickness), so a direct comparison of different actuators in this parameter does not make much sense. We give the strain value where the dimensions of the device are already taken into account, while in our work we pay special attention to the shape of the bending of the actuators. Since the formula for calculating the deformation according to the coordinates of the end of the actuator and its thickness for a bend that is different in shape from the arc of a circle is simply not correct.

The situation is the same with the generated force: its values depend on the size of the actuator. Therefore, in our opinion, it is much more correct to use the values of the generated mechanical stress, which does not depend on the dimensions. In the literature, the voltage is usually calculated from the modulus of elasticity of the electrodes. In this case, porous electrodes are an anisotropic material, and the elastic modulus is calculated under tension. Bending is described by both tension and compression, furthermore there is no guarantee that the modulus of elasticity of the actuator remains constant during its operation (due to Joule heating or for other reasons). We present the values of the mechanical stress, calculated according to the cantilever beam model, from the experimental data on the force as well as stress calculated from Yong modulus.

With ionic conductivity, too, not everything is so simple. In the case of pure electrolytes like EMIMBF4, ionic conductivity determines the rate of charge accumulation and, as a consequence, the rate of deformation of the actuator. In the case of membranes based on ionomers, different ions are present (H, imidazolium, 1-ethyl-3-methylimidazolium, etc.) and it is difficult to understand exactly how they are distributed. For example, in Terasawa, N. RSC Adv. 2017, 7, 2443–2449, doi: 10.1039/c6ra24925f. it was indicated that with an increase in ionic conductivity, the deformation of the actuator increases, and in Kim, O.; Shin, T.J.; Park, M.J. Nat. Commun. 2013, 4, 1–9, doi:10.1038/ncomms3208, on the contrary, an actuator based on a membrane with the lowest conductivity demonstrates maximum bending. In our work, the membrane with the highest deformation shows the best properties, while in other cases there is no correlation between conductivity, bending speed and deformation.

As a result, it is difficult to choose one most important parameter. We could compare each of the parameters with literature data, but we would have to select different articles for each parameter. We consider this approach not entirely fair, therefore we present a comparison only in the section with the results and do not put the "superiority" of new materials into the abstract and conclusion.

2. This question is closely related to the first point. Due to the lack of a unified methodology for characterizing actuators, we deliberately do not provide an overview of the properties of actuators. We agree that the absence of any characteristics does not allow the reader to evaluate our work, therefore, we indicated the average values of the deformation for the actuators described in the introduction.

3. In our work, for comparison, actuators based on PVDF without the addition of ionomers are manufactured and described; these devices have a "classical" composition of Bucky-gel actuators. We have also added to Tables 5 and 6 the strain and generated stress values for the previously described Nafion-based actuator. This actuator is the closest in structure and composition to the actuator described in our work.

4. We reread the text of the article for typos. We asked our English-speaking colleague to test the language

Reviewer 2 Report

The manuscript needs some minor revision

1) in section of 2.11, please provide the details on how to control the thickness of the membrane.

2) section of 2.12:  The authors mentioned the impedance was conducted at constant potential of 5mV. Please explain why the impedance was measured at constant potential of 5mV, not at the open circuit potential. Please specify the amplitude.

3)Section of 2.13: please explain why the sample was cut to 50mmX0.9mm. 0.9mm is probably too small.

4) Line 253-254: Samples of the same rectangular shape (~65x6 mm) of membrane and two electrodes were cut. This is confusing. Are the size of the membrane and electrode are same? If they have same size, the two electrodes may be in direct contact on the edges leading to the short circuit. (Then the actuators were cut to 60x5 mm). Please comment

Author Response

  1. The thickness of each membrane was measured with a micrometer. It is mentioned in section 2.12.
  2. Thank you for bringing this to our attention. You are right, this is a mistake in the description. The impedance was measured at an open circuit potential with an amplitude of 5 mV. A corresponding change has been made to the text of the manuscript.
  3. Here you are also right, the samples were cut with approximate dimensions of 50 by 9 mm (0.9 cm, not mm, of course). The exact width and thickness were measured with a caliper and micrometer, respectively, before testing for each specimen. We noted this in the text in section 2.12.
  4. Perhaps we have described the manufacturing process a bit confusingly. The membranes were cut out slightly larger (65x6 mm) than the electrodes (60x5 mm). After pressing, the actuators were cut to the size of the electrodes. As you rightly noted, this was done to avoid shorting the electrodes. We indicated the dimensions of the membranes and electrodes in the text of the article.

Reviewer 3 Report

The authors report different routes for the synthesis of poly(phenyl norbornenes) for applications in ionic actuators as polymer electrolytes, along with thorough characterization by different instrumental techniques. Interesting results are reported on membrane-based actuators outperforming the classical bucky gel actuator. The manuscript is well written and organized. I recommend it for publication after the authors add minor comments on the economic comparison of the membrane-based actuators and classical bucky gel actuators.

Author Response

Thank you for your positive feedback. Although we use the term "classical" to describe the first Bucky gel actuators based on ionic liquids, the topic of these actuators is fairly new. Most recent works, including ours, describe approaches to obtaining new ionomers and their applications. To give an economic assessment on the basis of laboratory technology or simply on the basis of the price of reagents, in our opinion, is premature. In addition, the cost of individual components, such as carbon nanotubes, Nafion or its analogues, depending on the supplier, may differ by several times or even by an order of magnitude. Therefore, carrying out an economic evaluation of ionic actuators is not such a simple task at the level of a separate full review, which is beyond the scope of this work.